# On Coordinate Decoding for Keypoint Estimation Tasks

## Reproducibility Summary

**Scope of Reproducibility**

A series of 2D (and 3D) keypoint estimation tasks are built upon ***heatmap*** coordinate representation, *i.e.* a probability map that allows for learnable and spatially aware encoding and decoding of keypoint coordinates on grids, even allowing for sub-pixel coordinate accuracy. In this report, we aim to reproduce the findings of DARK [1] that investigated the 2D heatmap representation by highlighting the importance of the encoding of the ground truth heatmap and the decoding of the predicted heatmap to keypoint coordinates. The authors claim that **a)** a more principled distribution-aware coordinate decoding method overcomes the limitations of the standard techniques widely used in the literature, and **b)**, that the reconstruction of heatmaps from ground-truth coordinates by generating accurate and continuous (non-quantized) heatmap distributions lead to unbiased model training, contrary to the standard coordinate encoding process that quantizes the keypoint coordinates on the resolution of the input image grid.

**Methodology**

To reproduce DARK, we thoroughly studied and followed the official implementation provided by the authors. We partially used the publicly available code and integrated it to a model development kit to accelerate the implementation and execution of our experiments. We conducted the experiments on a recent human pose and depth dataset aiming to validate the claimed concepts on different a different setting than the one originally used and assessed. Our experiments were conducted on an Nvidia 2080 Ti 12 GB GPU with an average training time of 20 hours.

**Results**

We were able to reproduce the findings of the original work. The proposed distribution-aware sub-pixel coordinate decoding improves the keypoint estimation performance, outperforming existing, standard decoding techniques. The elimination of coordinate quantization after re-scaling to the down-scaled input image resolution before the coordinate encoding (heatmap reconstruction), indeed leads to unbiased model training. Going a step beyond, we assess the performance of other sub-pixel decoding approaches, such as *Center of Mass* [2, 3], which also show increased performance.

**What was easy**

The DARK paper is well-structured and -presented aiding reproducibility. Further, the official code of the paper is publicly available, ensuring that the method and its results are reproducible. Given that DARK serves as a model-agnostic head that relies on post-inference processing only during testing, it is overall straightforward to implement and verify its effectiveness.

**What was difficult**

A potential issue has to do with the availability of new datasets, beyond the ones used in the paper, that provide coordinates in higher resolutions.

**Communication with original authors**

No communication was required, as the implementation was openly available and the verification is straightforward.

Submitted to ML Reproducibility Challenge 2020. Do not distribute.

# 1  Introduction

Keypoint coordinate estimation in $N-$dimensional grids is a common task related to a variety of computer vision domains such as human and object pose estimation, face tracking, crowd scene analysis and beyond. For modern data-driven methods, the heatmap representation, *i.e.* a $N$-dimensional probability distribution (Gaussian in most cases) centred at each keypoint coordinates, is considered the *de facto* coordinate representation for keypoint estimation. Heatmap representations enable the use of efficient fully $N$-dimensional convolutional networks, learning dense spatial features around the ground-truth location, and lending to a more natural way of estimation coordinates.

Despite the popularity of heatmap coordinate representation, DARK is the first research work that investigates heatmaps in depth, meaning that the authors explore and assess the standard *coordinate-to-heatmap encoding* and the *heatmap-to-coordinate decoding* techniques present in the literature, and formulate and assess a novel approach for distribution-aware coordinate representation.

# 2  Scope of reproducibility

We aim to reproduce the Distribution-Aware coordinate Representation of Keypoints (DARK) [1] method to assess its benefits in keypoint coordinate estimation. In particular, we reproduce and assess DARK in 2D human pose estimation with the use of state-of-the-art fully convolutional networks, such as *Stacked Hourglass* [4] and *HRNet* [5], similarly to the original work. A recent dataset, HUMAN4D [6], including multi-view depth data and 3D poses captured with the use of a high-quality VICON Motion Capture [6] setup, is used to train our models from scratch.

In a nutshell, the main claims of DARK paper, and those that we seek to validate, are summarised below:

- Claim #1: *Efficient Taylor-expansion based coordinate decoding improves coordinate estimation accuracy.*
- Claim #2: *Unbiased sub-pixel centred coordinate encoding results in optimal supervision and upgraded model performance.*

# 3  Methodology

Our approach in reproducing the work was to carefully re-implement every single step of the approach, following the presented methodology as well as studying the code of the official repository of the authors[1]. Beyond that, the code that we used as is from the authors' implementation is the Taylor-expansion based coordinate decoding. The rest of the implementation for all our experiments was developed in a model development kit [7], oriented towards reproduction.

The code and its documentation are submitted and published along with this report.

## 3.1  Models

Following the original paper, we selected and used the following models for predicting coordinate heatmap distributions in images:

- *Stacked Hourglass (SH) [4]*; with feature maps of 128 width and 4 stages and 18 output heatmap layers.
- *HRNet [5]*; with feature maps of 32 width and 4 stages and 18 output heatmap layers. The 2nd, 3rd and 4th stages contain 1 exchange block containing 4 residual units each.

## 3.2  Coordinate representation

### 3.2.1  Coordinate decoding

We consider a heatmap $\mathbf{H}^j$ that encodes the coordinates of each $j-$joint 2D position as predicted by our trained models. In our experiments, we decode the coordinate from $\mathbf{H}^j$ with $4$ different decoding methods. In the following description, $N_u$ and $N_v$ are the width and height of the heatmap, and $\mathbf{p} \in \Omega$ represents a pixel coordinate in the heatmap domain $\Omega$.

- *ArgMax*; by directly using the maximal activation location $m$ by:

$$m = (u,v)_j = \arg\max_{\mathbf{p} \in \Omega}(\mathbf{H}^j(\mathbf{p})) \tag{1}$$

---

[1] https://github.com/ilovepose/DarkPose/tree/612fad594eddd022b8a162a2c7274f9ee8d06d2c

- *Standard*; by slightly shifting (sub-pixel shifting) the estimated coordinates between the maximal $m$ and second maximal $s$ activation by:

$$p = (u, v)_j = m + 0.25 \cdot \frac{s - m}{||s - m||_2} \tag{2}$$

This method constitutes one of the standard coordinate decoding techniques that predicts the maximal activation with a greedy sub-pixel shifting (0.25 pixels) towards the second maximal activation in the heatmap space $\Omega$.

- *Taylor-expansion (DARK)*; exploring the distribution structure of the predicted heatmap to infer the underlying maximum activation assuming the predicted heatmap signal follows a 2D Gaussian distribution, similarly to the ground-truth heatmap during supervision by:

$$\mu = (u, v)_j = m - (\mathcal{D}''(m))^{-1} \mathcal{D}'(m) \tag{3}$$

where $\mathcal{D}'(m)$ and $\mathcal{D}''(m)$ denote the first and second derivative (i.e. Hessian) of the predicted heatmap after applying logarithmic transformation.

- *CoM*; considering heatmap $\mathbf{H}^j$ as a 2D grid of point masses determined by the corresponding heatmap pixel values, we calculate one single 2D point whose mass is the total mass of the grid by:

$$c = (u, v)_j = (\frac{1}{N_u}, \frac{1}{N_v}) \circ \sum_{\mathbf{p} \in \Omega} \mathbf{H}^j(\mathbf{p}) \cdot \mathbf{p} \tag{4}$$

where $\circ$ denotes element-wise multiplication. CoM constitutes by nature a sub-pixel coordinate decoding method that is not evaluated in the original paper.

### 3.2.2 Coordinate encoding

The second claim is about coordinate encoding where the existing standard methods down-sample the original input images into the model input resolution, quantizing the keypoint coordinates accordingly (*w/ quant*). Thus, the ground-truth heatmaps reconstructed based on the quantized coordinates are erroneous leading to sub-optimal supervision and degraded model performance.

Instead, DARK authors propose the heatmap generation placing the heatmap centre at the unbiased, non-quantized keypoint coordinate (*w/o quant*), claiming that, independently of the network architecture and the coordinate decoding method, the models perform consistently better.

## 3.3 Datasets

### 3.3.1 HUMAN4D

We investigate the benefits that DARK introduces in heatmap-based keypoint estimation by conducting our experiments on a recent dataset, HUMAN4D, that contains, among other data, pose annotated depth maps captured with the use of recent consumer-grade depth sensing devices. Beyond the experimentation on a dataset different from the ones used in the original work, the rationale behind this selection is twofold:

- The dataset provides the original 3D poses allowing for their projection as non-quantized 2D coordinates on the 2D depth map grids and, thereby, enabling the comparison between *w/ quant* and *w/o quant* coordinate encoding.
- The training and evaluation of baseline pose estimation data-driven models on depth data which are relatively limited in comparison with models trained on color data.

The activities included are the following: *running*, *jumping_jack*, *bending*, *punching_n_kicking*, *basketball_dribbling*, *laying_down*, *sitting_down*, *sitting_on_a_stool*, *talking*, *object_dropping_n_picking*, *stretching_n_talking*, *walking_n_talking*, *watching_scary_movie* and *in-flight_safety_announcement*.

### 3.3.2 Preparation

With the use of HUMAN4D 3D poses, we project the joint 3D positions to the 2D depth map grids to obtain the 2D non-quantized poses. Using the single-person part of the HUMAN4D dataset, clipping the first 100 frames from each sequence (to exclude T-Pose calibration frames) and sub-sampling the remaining part of the sequence, we create a set of $98,864$ single-view samples from $14$ single-person activities, including all $4$ depth maps per multi-view frame.

We use the data of the first three subjects performing all the activities for training, while we split the activities performed by the fourth subject (unseen) in two halves for validation and testing. We decide to split the data that way in order to assess the models on an unseen subject with different body structure, also keeping different activities between the validation and testing sets to provide fair conclusions with respect to the models' performance. The training, validation and testing data sets consist of $73, 248$, $12, 236$ and $13, 380$ samples, respectively.

Finally, beyond the application of center cropping on depth maps from $320 \times 180$ to $320 \times 160$ ($10px$ from top and bottom) to better fit to the multi-stage models, no further pre-processing steps were applied on the original data of the dataset.

## 3.4 Hyperparameters

All the models are trained for **30** epochs, picking for testing the best performing ones based on validation metrics across the epochs. Given that our aim is to assess the heatmap coordinate representation, not purely the effectiveness of the models in pose estimation, we use constant and common training hyperparameters for the various models, without further investigation for finetuning. To this end, both for *SH* and *HRNet*, we used the **Adam** optimizer with a learning rate of $2e - 3$, betas of values $0.9$ and $0.999$, without weight decay. The seed remained constant during all models' training, equal to 1314, ensuring reproducibility. We used simple MSE loss as in the original work that easily drove all models to convergence.

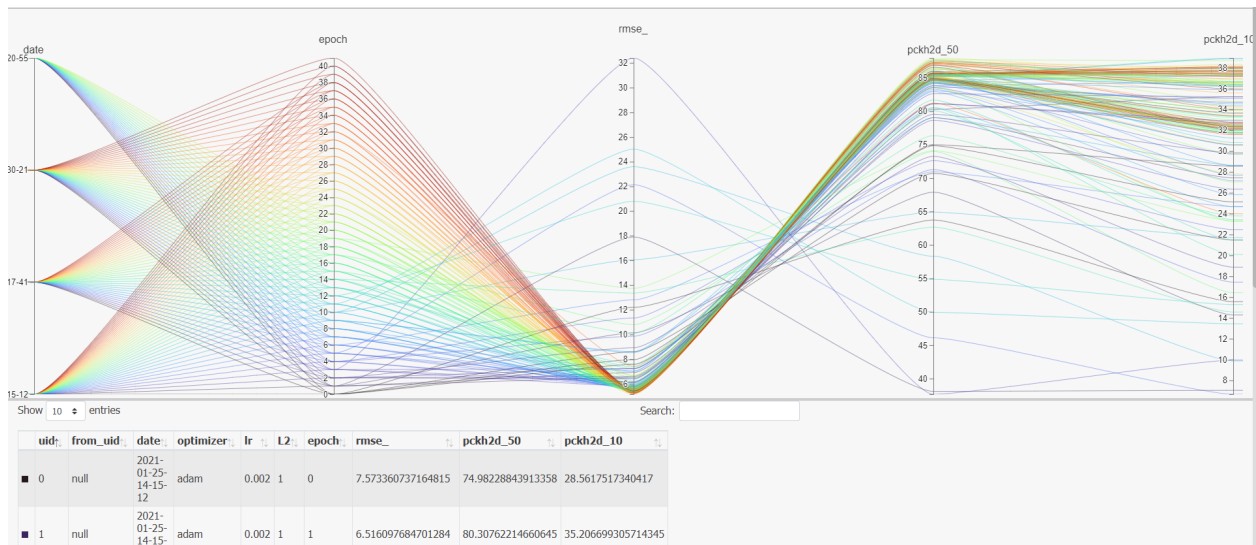

Figure 1: Interactive plots depicting the progress of the models' training and the validation metrics in the form of a local web page can be found in the supplementary material of the report.

## 3.5 Experimental setup and code

As already noted, carefully following the authors' description and code, we built our implementation upon *moai* [7], a PyTorch Model Development Kit for rapid development of Deep Learning workflows[2]. In this framework, we implemented our models as well as the heatmap coordinate encoding and decoding operations with the use of a configuration file that allows for the definition and configuration of the various components and hyperparameters.

Similarly to DARK paper, we report the following standard pose estimation metrics:

**RMSE**: represents the square root of the squared differences between estimated and ground truth keypoint coordinates in pixels.

**PCKh**: considers an estimated keypoint location correct only if the euclidean distance error is lower than a percentage threshold $\alpha$ of the euclidean distance between the head body part segment, i.e. from neck to head joint positions. In our experiments, we use *PCKh-0.1* and *PCKh-0.5* with $\alpha = 10\%$ and $\alpha = 50\%$, respectively.

---

[2]https://github.com/ai-in-motion/moai

## 3.6 Computational requirements

The training of the models was executed on a workstation machine with the hardware specifications figured in Table 1.

Table 1: Hardware Specifications

| | |
|---|---|
| *OS* | Windows Microsoft Pro (x64) |
| *Storage* | 3TB Toshiba HDD |
| *CPU* | Intel i9-7900X (4.30 GHz) |
| *GPU* | GeForce RTX 2080 Ti (12 GB) |
| *RAM* | 4 x 16 GB Kingston (2666 MHz) |

# 4 Results

Our experimental results validate both claims of DARK [1], as presented in Table 2. In particular, **w/o quant** models outperform the corresponding **w/ quant** ones, while *DARK* coordinate decoding method shows the best performance compared to the other competitive decoding methods.

## 4.1 Results reproducing original paper

The results presented in Table 2 support both claims of the paper, allowing the reader to easily study and relatively assess the coordinate decoding and encoding achievements of each method.

Table 2: The overall *PCKh-0.1*, *PCKh-0.5* and *RMSE* results of **SH_4S** and **HRNet_4S** on HUMAN4D trained **w/ quant** and **w/o quant** coordinate encoding methods. The final keypoint estimation is assessed with the use of various coordinate decoding methods.

| Model + *Coordinate Decoding* | PCKh-0.1 ↑ | PCKh-0.5 ↑ | RMSE ↓ |
|---|---|---|---|
| **SH_4S** (w/ quant) + *ArgMax* | 35.38 | 68.14 | 4.12 |
| **SH_4S** (w/o quant) + *ArgMax* | 39.80 | 68.78 | 3.75 |
| **SH_4S** (w/ quant) + *Standard* | 36.37 | 68.36 | 4.07 |
| **SH_4S** (w/o quant) + *Standard* | 41.18 | 69.05 | 3.69 |
| **SH_4S** (w/ quant) + *CoM* | 37.77 | 70.35 | 3.87 |
| **SH_4S** (w/o quant) + *CoM* | 44.24 | 71.45 | **3.33** |
| **SH_4S** (w/ quant) + *DARK* | 39.54 | 70.75 | 3.87 |
| **SH_4S** (w/o quant) + *DARK* | **45.17** | **71.65** | 3.47 |
| **HRNet_4S** (w/ quant) + *ArgMax* | 37.29 | 68.10 | 3.93 |
| **HRNet_4S** (w/o quant) + *ArgMax* | 39.19 | 68.60 | 3.85 |
| **HRNet_4S** (w/ quant) + *Standard* | 38.48 | 68.35 | 3.87 |
| **HRNet_4S** (w/o quant) + *Standard* | 40.50 | 68.90 | 3.79 |
| **HRNet_4S** (w/ quant) + *CoM* | 33.95 | 68.38 | 3.94 |
| **HRNet_4S** (w/o quant) + *CoM* | 36.77 | 70.21 | 3.79 |
| **HRNet_4S** (w/ quant) + *DARK* | 41.73 | 70.29 | 3.65 |
| **HRNet_4S** (w/o quant) + *DARK* | **44.83** | **71.52** | **3.55** |

As mentioned above, our main aim is to highlight the effectiveness of the distribution-aware heatmap representation, instead of assessing the performance and comparison between the models. Nevertheless, in our experiments, a general comment could be that **SH_4S** performs better than **HRNet_4S** almost in all combinations.

### 4.1.1 Taylor-expansion coordinate decoding - Claim #1

Assessing the various coordinate decoding methods we implemented and evaluated, as presented in Sec. 3.2.1, Claim #1 is fully supported since the Taylor-expansion coordinate decoding does improve the coordinate estimation performance. Considering the predicted heatmap as a Gaussian heatmap and applying Taylor-expansion coordinate decoding shows

remarkable results against the rest of the methods. In particular, as shown in Table 2, DARK decoding performs better in all experiments for all metrics, with the only exception the RMSE error in "*SH_4S (w/o quant) + CoM*" experiment. In that case, DARK is more accurate than CoM, but CoM exhibits a smaller average error. This means that DARK suffers from larger outliers than CoM, a partly reasonable case given their nature. CoM uses the entire heatmap to reach a consensus about the coordinate, while DARK uses local heatmap statistics around the maximum activation, but is more susceptible to heatmap noise. When applying DARK, a low-pass Gaussian filter is first used, which is standard practise before the application of derivative extraction, in order to suppress noise and capture only the signal's high frequency information. However, this might not always be the case in out-of-distribution data predictions. When the predictions follow the underlying assumptions of DARK, namely a 2D Gaussian distribution, its finer-grained localisation will be effective. In the case of distribution noise, the globalised nature of CoM will be more robust. Evidently, the SH experiment is one such case, but nonetheless, DARK's accuracy indicates that it still offers favourable behaviour.

### 4.1.2 Non-quantized coordinate encoding - Claim #2

Observing again the results in Table 2 from another perspective, we can assess the validity of Claim #2, i.e. that the coordinate quantization before heatmap generation results in inaccurate and biased ground-truth signals that reduce the model performance. Indeed, for both architectures, *SH* and *HRNet*, and for every coordinate decoding method, *w/o quant* models show higher *PCKh* accuracy and lower *RMSE* errors, totally supporting the second claim of the authors.

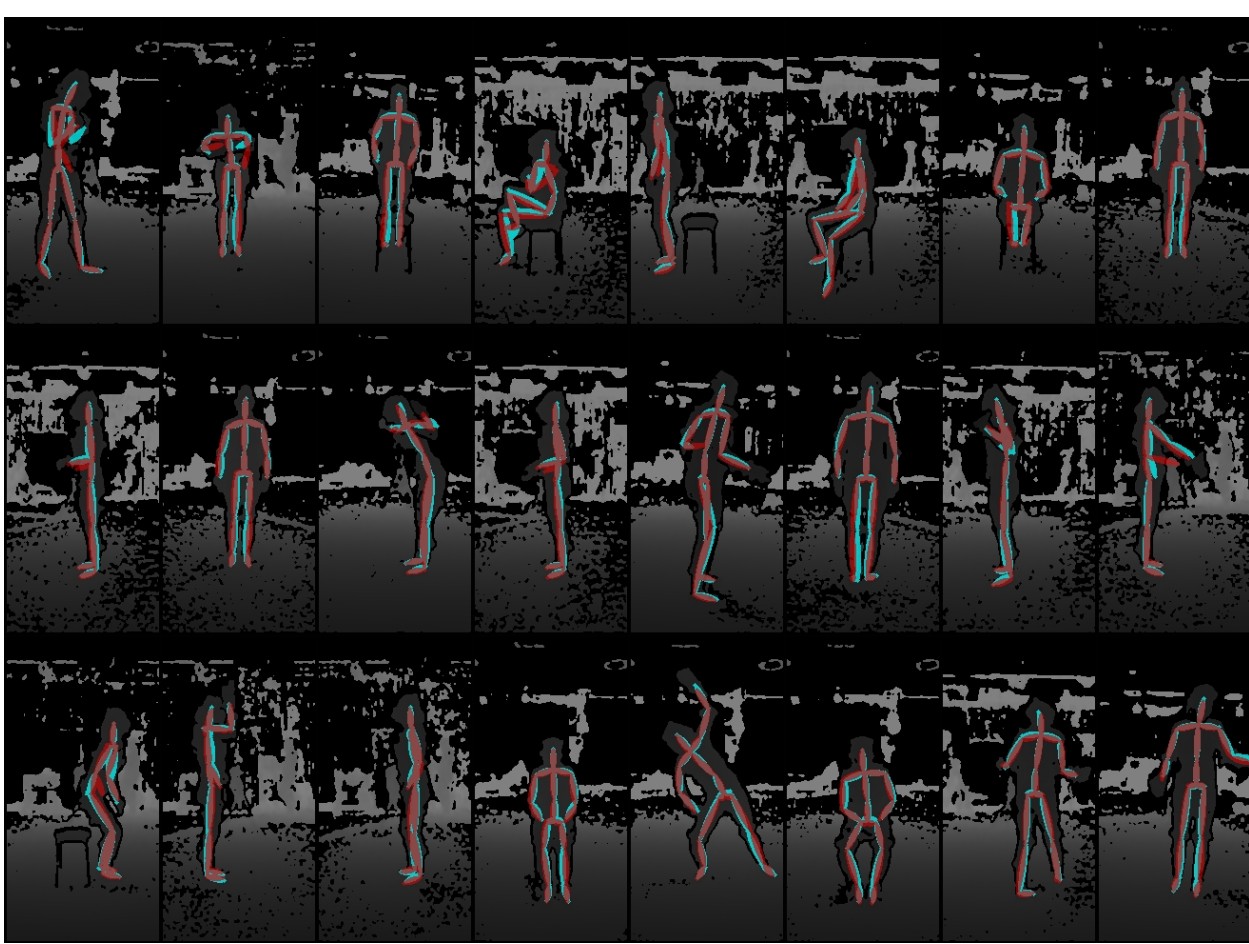

Figure 2: Pose estimation qualitative results with the use of DARK heatmap decoding on HUMAN4D testing dataset (unseen subject). Cyan and red colours indicate the ground-truth and the predicted poses, respectively.

## 4.2 Results beyond original paper

An experiment we conducted, beyond the experiments of the present paper, is the comparison between DARK and spatial coordinate regression methods for keypoint estimation, such as Center of Mass (CoM) [2, 3].

### 4.2.1 DARK against Center of Mass

Observing Table 2, DARK obviously outperforms CoM almost in all experiments. Nevertheless, it is worth-noting that focusing on the results of the best performing model, as shown in Table 3, CoM demonstrates comparably to DARK. Especially in RMSE, which is a continuous metric, CoM presents lower errors than DARK, while, generally, its performance is higher when decoding heatmaps predicted by **SH_4S**.

Table 3: *PCKh-0.1*, *PCKh-0.5* and *RMSE* results of best performing model (**SH_4S** (w/o quant)) with the use of DARK and CoM coordinate decoding methods.

| Model + *Coordinate Decoding* | PCKh-0.1 ↑ | PCKh-0.5 ↑ | RMSE ↓ |
|---|---|---|---|
| **SH_4S** (w/o quant) + *CoM* | 44.24 | 71.45 | **3.33** |
| **SH_4S** (w/o quant) + *DARK* | **45.17** | **71.65** | 3.47 |

## 5 Discussion

Summarizing the present reproduction study, we conclude that the DARK paper introduces a novel, simple, lightweight and highly effective method for heatmap coordinate encoding/decoding that boosts the keypoint estimation accuracy.

DARK outperforms standard greedy sub-pixel coordinate decoding methods. Furthermore, it shows higher accuracy than spatial coordinate regression methods, such as CoM, which also shows lower erroneous predictions than standard methods. On top of those, DARK can be considered a significant improvement for multi-person pose estimation. That is due to the fact that spatial coordinate regression methods, though highly effective and fully differentiable, struggle in the regression of more than one keypoint location when more than one heatmap in the same heatmap layer, i.e. specific joint, are present in the same predicted layer, a common case in multi-person datasets.

With regards to identified gaps in the present heatmap representation study, we believe that spatial coordinate regression methods, such as CoM or MoM (median of mass) [8], should have been taken into account, benchmarked and reported in this paper. Several recent heatmap-based keypoint estimation data-driven methods have been developed with the use of spatial coordinate regression, which beyond its effectiveness, allows the design of end-to-end, fully differentiable architectures with direct supervision on the actual task of keypoint estimation, i.e. the coordinate distance error between the predictions and ground-truth.

### 5.1 What was easy

Implementing most of the code was straightforward as authors provide their source code on GitHub. Apart from that, past GitHub issues were another source of retrieving information, clarifying parts of the paper when needed.

### 5.2 What was difficult

All and all, this reproduction study was smooth without bottlenecks, delays or other barriers. A potential difficulty that can be considered was the availability of human pose datasets, beyond the ones used in the paper, that provide coordinates in higher resolutions, in order to further investigate the quantization error effects.

### 5.3 Communication with original authors

As implementing the code of the paper was straightforward and the ideas presented in the paper are clear and well-defined, we did not communicate with the authors.

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
