# OpenReview forum: "On Coordinate Decoding for Keypoint Estimation Tasks"
_ML_Reproducibility_Challenge/2020 — Reject_

### Official Review · AnonReviewer3 · 2021-03-01
**A solid replication, the results of the comparison leave room for interpretation.**

**Rating:** 7
**Confidence:** 3

**Review:**

The manuscript presents a replication study on the paper "Distribution-aware coordinate representation for human pose estimation"

Pros:
The authors are able to replicate the method, largely because of the available code from the original paper.
The authors tested on an independent dataset. They also introduced additional comparison with other models. Both seem to be extension and qualitatively support the claims of the original paper.
The authors were able to successfully implement both the proposed algorithms from the description of the algorithms in the original paper (they didn’t need to contact thus with the original authors for testing reproducibility).

Cons:
The missing of validation on the COCO dataset used by the original paper. This weakens the successful replication claim.
The authors implementation is not in a github repository


**Familiar With The Original Paper:**

I have read the original paper

**Reproducibility Summary:**

Report has summary

---

### Official Review · AnonReviewer4 · 2021-03-14
**Concise well executed report, lacks discussions**

**Rating:** 6
**Confidence:** 4

**Review:**

* Reproducibility Summary

  The report presents a well-written, concise reproduction study on the paper DARK. The report contains a reproducibility summary that highlights the scope, methodology, results, and what was easy/difficult appropriately as required by the challenge.

* Scope of reproducibility

  The report investigates two central claims of the original paper.

* Code: whether reproduced from scratch or re-used author repository.

  Authors re-use the repository of the original papers for most of their experiments. They also implement their own code in a novel reproducible model development kit following the author's code. I find this reproducibility kit (moai) fascinating, and a great example of a submission that introduces a reproducible paradigm to test original authors' codes.

* Communication with original authors

  The report mentions they did not communicate their results/findings with the original authors.

* Hyperparameter Search

  The authors seem to investigate a modest subset of hyperparameters for their work.


* Ablation Study

  The authors perform an ablation study on the originally proposed algorithm by evaluating the claims on a new dataset, HUMAN4D. This kind of robustness evaluation on a new dataset is very much welcome, as it adds valuable insights to the proposed algorithms.

* Discussion on results

  The report contains a limited discussion of the results on the HUMAN4D dataset using DARK. For claim 1, the authors note DARK decoding performs better for all experiments except one. It would have been better to add a discussion of why this exception occurs. The report also extends the original results by adding a new experiment with comparing with CoM. The report can be made stronger by exploring more ablations to shed light on the effectiveness of DARK, and/or adding more discussion to the effectiveness of DARK.

* Recommendations for reproducibility

  The authors highly commend the original paper on their state of reproducibility.

* Overall organization and clarity

  The paper is well organized and well written.

**Familiar With The Original Paper:**

I have not read the original paper

**Reproducibility Summary:**

Report has summary

---

### Decision · Program_Chairs · 2021-03-31

**Decision:**

Reject

**Comment:**

Overall reviews and/or the paper content not good enough for the AC to recommend to the journal.